# Preference Heterogeneity of Coastal Gray, Green, and Hybrid Infrastructure against Sea-Level Rise: A Choice Experiment Application in Japan

**Yui Omori**

Division of Natural Resource Economics, Graduate School of Agriculture, Kyoto University, Oiwake-cho, Kitashirakawa, Sakyo-ku, Kyoto 606-8502, Japan; omori.yui.27x@st.kyoto-u.ac.jp

**Abstract:** Coastal zones are bearing the brunt of an increase in the likelihood of extreme events, coupled with sea-level rise (SLR). Conventionally, gray infrastructures, such as seawalls, have been constructed to reduce risks in limited coastal zone spaces. Nature-based approaches, known as green infrastructure, have been used in coastal defense, and their ecosystem-based disaster risk reduction functions (Eco-DRR) have received growing attention. However, both gray and green infrastructure alone have limitations in responding to an ongoing increase in the intensity and frequency of natural hazards. To overcome these issues, hybrid infrastructure, which combine gray and green components, is needed, and they have been receiving growing attention. Meanwhile, a large-scale coastal development requires stakeholder agreement; thus, it is imperative to understand people's demands and build a consensus between municipalities and coastal citizens in coastal development for long-term resilience. The author administered the online survey across Japan, applying it to the choice experiment, and obtained 840 valid responses. Therefore, this paper clarified the heterogeneities in coastal people's preferences for coastal ecosystem services provided by gray, green, and hybrid structures in intertidal zones in Japan, recognizing seawalls as gray and coastal pine forests as green infrastructure. Consequently, while coastal citizens acknowledged gray's coastal defense function, the diverse perceptions toward seawalls for SLR preparation were notable as its scenarios became severe. Another remarkable finding is that nearly 60% of respondents preferred Eco-DRR functions provided by coastal forests with JPY 695 in willingness-to-pay for expanding 100 m in width, even though there are uncertainties in their performances.

**Keywords:** gray infrastructure; green infrastructure; hybrid infrastructure; seawalls; coastal forests; sea-level rise; choice experiments; mixed logit model

## 1. Introduction

Coastal areas are often subject to extreme events such as storm surges and typhoons [1]. Sea-level rise (SLR) is likely to increase these risks [2,3], making coastal citizens in lowlands vulnerable to coastal hazards [4]. A rising of 10 to 20 cm of the sea level is expected no later than 2050, which will more than double the frequency of extreme water-level events, especially in low-lying coastal cities and the habitable parts of Pacific Island nations [5]. Meanwhile, despite the growing concerns about climate change coupled with SLR, population growth in coastal zones is accelerating, and more than half of coastal countries have 80–100% of their population within 100 km of their coastlines [6]. A global analysis of human settlements in coastal areas revealed that they are concentrated within 5 km of coasts and that the average population densities are higher at elevations below 20 m and 100 km in width from the coastline [7]. This trend can be seen in studies on coastal megacities that are severely exposed to ecological and anthropogenic drivers [8–10]. Their spatial advantages have led to the development of human settlements and tourism, as coastlines provide resources, trading, and job opportunities. Meanwhile, the vulnerability of coastal zones requires them to be resilient [11].

Given population growth and increasing threats of natural hazards in coastal zones, the resilience of coastal communities—the ability of socio-ecological systems to absorb disturbances and recognize, while undergoing change, so as to still retain essentially the same functions and structures ([12–14])—is more critical than ever [15]. Conventionally, artificial structures (termed gray infrastructure, built infrastructure, or hardened structures, encompassing seawalls, levees, culverts, and bulkheads) have been constructed to reduce risks in limited coastal zone spaces. In particular, seawalls are considered to be the last line of defense as they are crucial in stabilizing the shoreline, and thus essential for ensuring the safety of residential livelihood [16].

Nature-based approaches, known as green infrastructure, have been used in coastal defense, and their ecosystem-based disaster risk reduction functions (Eco-DRR), which ecosystems serve as disaster mitigation, have received growing attention. Green infrastructure, such as coral reefs, saltmarshes, sand dunes, mangroves, and coastal forests, not only functions as Eco-DRR, but also play a significant role in offering multiple co-benefits such as recreation, aesthetic values in landscape, food resources, and habitats for fauna and flora [17]. In recent years, the concept of green infrastructure has been applied to coastal infrastructural management, and research on coastal green infrastructure for adapting to climate change has accelerated [18]. Potential types of coastal infrastructure for coastal protection have been identified [19–21]. However, both gray and green infrastructure on their own have limitations in responding to the ongoing increase in intensity and frequency of natural hazards [22]. Generally, gray infrastructure is designed for natural hazards of a certain magnitude, ensuring greater protection within a capacity [23]. In other words, artificial defense structures could not function if the intensities of natural hazards were beyond their thresholds.

Coastal engineering has often been criticized for focusing on disaster reduction functions against unexpected events while paying little attention to coastal environments ([24]), in addition to higher construction and maintenance costs in the long term and reduced life expectancy of existing built structures [25,26]. Moreover, hard structures could worsen the loss of sandy beaches due to sea-level rise, exacerbating the future impacts of coastal hazards [27]. Meanwhile, green infrastructure is expected to keep pace with SLR and adapt to unexpected events [25,28]. Due to the growing awareness of the co-benefits and the cost-effectiveness in ecosystem services (ESSs) offered by green infrastructure, an economic analysis of this evidence is available [29]. For example, the value of coastal wetlands for ecosystem protection against hurricanes in the United States was estimated, and the potential storm protection was calculated to be USD 23.2 billion annually using a regression model [30]. However, uncertainties regarding their Eco-DRR function remains and nature-based disaster reduction has been called into question considering the future climate.

Some studies have mentioned that coastal vegetation is suitable for controlling sedimentary dynamics in response to gradual phenomena such as SLR, but it does not directly reduce erosion [31]. Other coastal green components, such as mangrove forests, can accrete sediments, leading to wave attenuation; however, it is still challenging to determine whether they can keep pace with the SLR [32,33]. In addition, vegetation in wetlands has an influence on land formation, accumulation of organic matter, and soil volume, enhancing the resistance to erosion, but it would be necessary to monitor vegetation changes and climate warming [34]. Thus, both coastal gray and green infrastructure need to consider several future scenarios and unforeseen events.

To tackle these issues, hybrid infrastructure, which combines gray with green components, has been received growing attention [25]. Even though only a few studies have focused on hybrid infrastructure, ecological engineering, which is a relatively new discipline that combines engineering and ecology, has emerged, and various studies regarding gray to green regime shifts have been conducted [19,35–39]. In addition, gray and green integrated approaches, such as mangroves and dykes [40], and their effectiveness for peak

water level attenuation under the circumstances of storm tides offered by marshes and dykes have been analyzed [41].

Owing to the numerous options to combine gray–green infrastructure and the uncertainties of its disaster risk reduction function against SLR, evaluating the economic values of hybrid infrastructure is difficult. This financial challenge is an ongoing issue, which has led to limited implementations and data regarding hybrid infrastructure. Furthermore, large-scale concrete structures have been criticized for their harmful impacts on the coastal environment [42]; thus, understanding people's demands and building consensus between municipalities and coastal citizens in coastal development for long-term resilience planning is becoming increasingly imperative [43,44]. A study conducted in New Jersey found that in coastal people's perceptions of coastal infrastructure, nature-based approaches (i.e., wetlands and dunes) are preferable to gray infrastructure (revetments and groins) [45]. In particular, people in Japan who frequently visit the sea prefer to conserve the shoreline, which provides multiple ESSs [46].

Overall, coastal infrastructure projects that combines gray and green infrastructures is always challenging not only due to a lack of scientific evidence regarding hybrid approaches but also due to the scarcity of data on what coastal design and planning is preferable for users.

Therefore, this study focuses on: (1) visualizing coastal people's preferences for coastal ESSs provided by gray, green, and hybrid infrastructures in the intertidal zones, and (2) exploring their decision making under the uncertainties in future SLR scenarios using choice experiments.

## 2. Materials and Methods

### 2.1. Study Site

The study site, Japan, has traditionally developed various ways for co-existing with natural events such as typhoons, storm surge and tsunamis. For example, seawalls are often used to maintain residential livelihoods, as they play important roles in stabilizing the shoreline and protecting the coastal communities of Japan. *Pinus thunbergii* (black pine trees) have been traditionally used as a nature-based disaster mitigation method to collect blown sand, mitigate wind speeds, and protect agricultural products and residential buildings [47]. Due to the economic advantages typically ascribed to coastal agglomeration, Japan has the largest coastal urban megacities (i.e., Tokyo, Osaka–Kobe, and Nagoya) [6,10], which often leads to a reliance on gray-based coastal defense that enables implementation in limited spaces. However, coastal design for disaster management was urged following the 2011 earthquakes and tsunamis that struck Japan's coastal communities. Therefore, while considering coastal disaster risk reduction management for resisting further increases in extreme events, it is important to understand that SLR is inevitable. For clarity regarding coastal environments in Japan, this study focused on intertidal zones, and seawalls and coastal pine forests were defined as coastal gray and green infrastructures, respectively (hybrid approach: seawalls and coastal forests). Figure 1 overviews gray (seawalls), green (coastal pine forests), and hybrid infrastructures (gray and green), and Table 1 summarizes their strengths and weaknesses.

### 2.2. Data Collections

In this project, an online survey intended for people in their 20s to 60s in Japan was administered by Nikkei Research Inc. between 6 March and 10 March 2020. The subjects were randomly selected and 959 responses were obtained (see the summary in Table 2). Note that the distances and elevations listed in Table 2 were calculated with the open-source Geographic Information System QGIS 3.10 using postal codes. Further, answering postal codes was optional, and responders could type specific numbers in the questionnaires if they did not wish to provide a response. There were 840 valid responses. The sample distribution is shown in Figure 2.

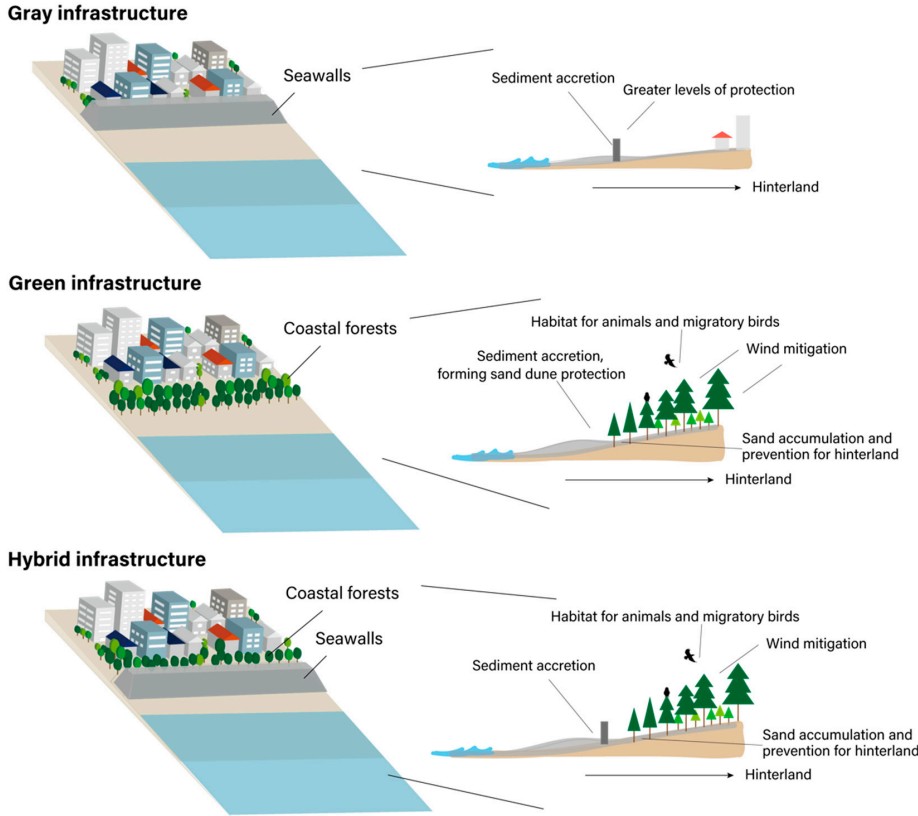

**Figure 1.** Gray, green, and hybrid infrastructure.

**Table 1.** Summary of strengths and weaknesses.

| Infrastructure | Strengths | Weaknesses |
|---|---|---|
| Gray<br><br>(seawalls) | Greater level of protection:<br><br>- alleviates speed of waves, tsunamis<br><br>- prevents erosion<br>- withstands storm events soon after seawalls are built and provides stable protection for two to three decades<br>Greater understanding of techniques and effects:<br>- allows policy makers to perform cost-benefit analysis<br><br>Significant engineering expertise. | Coastal habitat loss:<br>- has negative effects on other ecosystem services, in which coasts and surrounding areas such as beaches provide<br>High installation and maintenance cost:<br><br>- requires significant studies and additional elements to mitigate the effects of SLR<br><br>- weakens with time<br><br>Lack of community involvement:<br><br>- leads to safety misunderstandings and disaster risks |
| Green<br>(coastal forests) | Co-benefits of:<br><br>- coastal protection<br><br>- aesthetic seascape<br>- recreational use<br><br>- coastal habitats with many species<br><br>Lower cost<br>Adaptation to unexpected events:<br>- may keep pace with climate change, sea-level rise | Ambiguous effects:<br>- limited understanding regarding protection levels because of<br>topography, vegetation, seasons, and soils<br>Time for mature forests:<br>- requires approximately 20 years to mature for sufficient protection.<br>Pine wilt disease:<br>- damaged by diseases and pests<br>Other societal disadvantages (crime, dumping):<br>- requires appropriate maintenance |
| Hybrid<br><br>(gray and green: seawalls and forests) | Greater protection with other co-benefits:<br>- may require less space than natural approaches alone<br><br>Innovative coastal design and planning:<br><br>- compatible with resilience and authentic value | Little data and limited expertise:<br><br>- requires more research regarding potential effects<br><br>- may require more space to introduce both systems |

**Table 2.** Data summary.

| Gender | N | Proportion | Distance (from the Coastline: km) | N | Proportion | Frequency of Coastal Use | N | Proportion |
|---|---|---|---|---|---|---|---|---|
| Female | 491 | 51.2% | <5 | 253 | 26.4% | Almost every day | 11 | 1.1% |
| Male | 467 | 48.7% | 5–10 | 179 | 18.7% | 3–5 times/week | 11 | 1.1% |
| Others | 1 | 0.1% | 10–15 | 110 | 11.5% | 1–2 times/week | 18 | 1.9% |
| | | | 15–20 | 65 | 6.8% | 1–2 times/month | 59 | 6.2% |
| **Age** | | | 20–30 | 94 | 9.8% | 1–2 times/year | 120 | 12.5% |
| 20s | 154 | 16.1% | 30–40 | 50 | 5.2% | Vacation use | 107 | 11.2% |
| 30s | 193 | 20.1% | 40–50 | 37 | 3.9% | Seldom | 358 | 37.3% |
| 40s | 204 | 21.3% | >50 | 52 | 5.4% | None | 263 | 27.4% |
| 50s | 180 | 18.8% | Missing data | 119 | 12.4% | Others | 12 | 1.3% |
| 60s | 228 | 23.8% | Minimum value | 0.1 | | | | |
| | | | Maximum value | 107.1 | | | | |
| | **Income (million JPY)** | | **Elevation (m)** | | | | | |
| <2 | 102 | 10.6% | <5 | 188 | 19.6% | | | |
| 2–4 | 207 | 21.6% | 5–10 | 86 | 9.0% | | | |
| 4–6 | 224 | 23.4% | 10–15 | 72 | 7.5% | | | |
| 6–8 | 156 | 16.3% | 15–20 | 56 | 5.8% | | | |
| 8–10 | 124 | 12.9% | 20–30 | 86 | 9.0% | | | |
| 10–12 | 52 | 5.4% | 30–40 | 67 | 7.0% | | | |
| 12–14 | 25 | 2.6% | 40–50 | 48 | 5.0% | | | |
| 14–16 | 26 | 2.7% | 50–100 | 141 | 14.7% | | | |
| 16–18 | 6 | 0.6% | >100 | 96 | 10.0% | | | |
| 18–20 | 11 | 1.1% | Missing data | 119 | 12.4% | | | |
| 20–22 | 8 | 0.8% | Minimum value | −1.8 | | | | |
| >22 | 13 | 1.4% | Maximum value | 930.6 | | | | |
| Missing data | 5 | 0.5% | | | | | | |

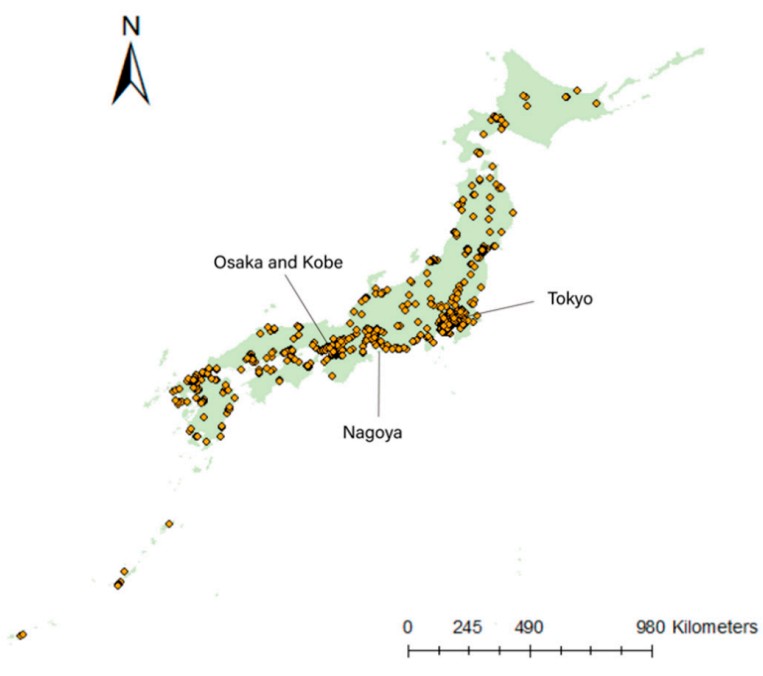

**Figure 2.** The sample distribution across Japan.

*2.3. Methods*

2.3.1. Experimental Design

This study applied choice experiments, which is one of the evaluation tools for valuing intangible goods and services such as ESSs. There are several methods to estimate non-market values, including travel cost or hedonic pricing methods, which are known as revealed preferences. However, these methods are not applicable to marine ESS evaluations because of their far-reaching direct and indirect effects [4,48]. Because of this, stated preference methods, which ask people questions about hypothetical situations, were

developed [49]. Choice experiments, which are one of the stated preference methods, allow researchers to consider the tradeoffs among ESSs [50,51]. Thus, choice experiments have been recently applied to coastal research [46,52], providing information regarding some changes in coastal settings and asking their most preferred options. Following this trend, this project applied choice experiments to clarify people's preferences toward coastal ESSs and their tradeoffs by jointly considering several important attributes for different hypothetical coastal settings (Figure 3).

| | Plan 1 | Plan 2 | Plan 3 | Plan 4 |
|---|---|---|---|---|
| Additional seawall height | the preparation for SLR: 0.5m ~ less than 1.0m | status quo | the preparation for SLR: 1.0m ~ less than 3.0m | Choose nothing |
| Forest width | 0 m | 500 m | 300 m | |
| Landscape | seawall | coastal forest | both | |
| Recreation | No recreational services | Camping & Walking | Camping, Walking & Fishing | |
| Bird species | 3 kinds of birds | 10 kinds of birds | 20 kinds of birds | |
| Annual tax (JPY) | 30,000 | 3000 | 10,000 | |
| Most preferable | ○ | ○ | ◉ | ○ |

**Figure 3.** Example of question.

The attributes and levels used in the choice experiments are listed in Table 3. To set the levels, RCP 2.6 and RCP 8.5, as the expected SLR of 0.43 m (ranged 0.29 m–0.59 m) and 0.84 m (ranged 0.61 m–1.10 m) by 2100 were considered [53]. Then, this study assumed that these scenarios were applied to all sea levels in Japan, and that seawalls with a height of 2 m and 2000 m were already there. The detailed environmental settings were explained before the choice experiment (Figures A1–A5), and it was iterated eight times for each respondent, during which the levels of attributes changed. The contributions are that the additional seawall height and coastal forest width are associated with gray and green components, respectively, which can influence people's perceptions of the disaster risk reduction function offered by gray and green infrastructure under the uncertainties in future scenarios and their effects. Furthermore, inserting gray and green attributes could help provide a picture of hybrid approaches and the extent to which hardened and natural components can be integrated.

**Table 3.** Attributes and levels of coastal settings.

| Attributes | Levels |
|---|---|
| Additional seawalls height (security) | ±0/ +1 m–+2 m (SLR: 0.5 m~less than 1.0 m)/ +2 m–+5 m (SLR: 1.0 m~less than 3.0 m)/ + over 5 m (SLR: 3.0 m~less than 5.0 m) |
| Forest width | 0/100 m/200 m/300 m/500 m |
| Landscape | Coastal forest only/Seawall only/Both |
| Coastal recreation | Walking only/Camping and Walking/Fishing only/Camping, Walking and Fishing/Nothing |
| Coastal biodiversity (bird species) | 3/10/20 kinds of birds |
| Annual tax (JPY) | 1000/3000/5000/10,000/30,000 |

### 2.3.2. Econometric Models

Conditional Logit (CL) and Mixed Logit (ML) Models

Discrete choice models are derived based on utility maximization in which the decision maker selects an alternative that offers the greatest utility. Let $U_{ni}$ denote the utility that respondent $n$ obtains from alternative $i$ in choice set $C_n$ as follows:

$$U_{ni} = V_{ni} + \varepsilon_{ni}, \tag{1}$$

where $V_{ni}$ is a deterministic component and $\varepsilon_{ni}$ is a random component, which are both assumed to be known to the individual but unknown to the analyst. The probability that respondent $n$ will choose $i$ from alternative $j$ in choice set $C$ is the probability that $U_{ni}$ is larger than $U_{nj}$, and is as follows:

$$\Pr(i) = \Pr[U_i > U_j] = \Pr[V_i - V_j > \varepsilon_j - \varepsilon_i] \forall j \neq i, \ \forall j \in C, \tag{2}$$

where the index $n$ is omitted for simplification. Note that $U_{ni}$ depends on parameters that are unknown to the researcher, and $\varepsilon_{ni}$ is the unobservable portion that respondent $n$ chooses over alternative $i$.

This probability is a cumulative distribution:

$$Pr_i = \Pr(\varepsilon_j - \varepsilon_i < V_i - V_j \ \forall j \neq i) = \int I(\varepsilon_j - \varepsilon_i < V_i - V_j \ \forall j \neq i) f(\varepsilon) d\varepsilon, \tag{3}$$

where $I(\ ;)$ is an indicator function. Random utility models are obtained from different density specifications, which are described as follows:

$$U_{ni} = \beta' x_{ni} + \varepsilon_{ni}. \tag{4}$$

When the utility is linear in $\beta$ and $x_{ni}$ is a vector of explanatory variables that are observed by the analysts and encompass the alternative attributes in a choice task, then $V_{ni}(\beta) = \beta' x_{ni}$.

The distribution of the random component $(\varepsilon_i)$ is assumed to be a type I extreme value, and the probability that responder n chooses alternative $i$ can be described in the conditional logit (CL) model and is expressed as follows [54]:

$$P_{ni} = \frac{\exp(\beta' x_{ni})}{\sum_{j=1}^{J} \exp(\beta' x_{nj})}, \tag{5}$$

where the preference parameter is assumed to be constant for all respondents [55]. When assuming the heterogeneity of preferences that vary across individuals, mixed logit (ML) probabilities are the integrals of standard logit probabilities over density parameters in the following equation:

$$P_{ni} = \int L_{ni}(\beta) f(\beta|\theta) d\beta, \tag{6}$$

where $L_{ni}(\beta)$ is the logit probability evaluated at $\beta$, and $f(\beta|\theta)$ is a density function, in which $\theta$ refers to the distribution. Thus, the ML probability takes the following form:

$$P_{ni} = \int \left( \frac{e^{\beta' x_{ni}}}{\sum_j e^{\beta' x_{nj}}} \right) f(\beta) d\beta. \tag{7}$$

ML is a mixture of logit functions that evaluate different $\beta'$s with $f(\beta)$ as the mixing distribution, wherein the utility (Equation (4)) takes a coefficient form [55], in which each of the coefficients is given an independent normal distribution with an estimated mean and standard deviation.

Estimation in ML Model

The log likelihood for the model is described as follows:

$$LL(\theta) = \sum_n lnP_{ni}(\theta). \tag{8}$$

Then, Equation (8) is approximated by simulating for any given $\theta$, in which the simulated log likelihood is determined by maximizing $LL(\theta)$. Thus, Equation (8) can be rewritten as follows:

$$SLL(\theta) = \sum_{n=1}^{N} \sum_{j}^{J} d_{nj} ln \frac{1}{R} \sum_{r=1}^{R} L_{ni}(\beta^r), \tag{9}$$

where $R$ is the number of draws and $d_{nj} = 1$ if responder n chooses $j$ plan and $d_{nj} = 0$ if they do not. In the context of discrete choice models [55], the superior coverage of Halton draws and its effectiveness compared with random draws has been explained, stating that 100 Halton draws improve model accuracy. Thus, this study utilized the maximum simulated likelihood with 100 Halton draws applied to the ML model.

Willingness to Pay (WTP)

For analysis, this article first elucidated the heterogeneities in preference using the ML model. Comparing CL and ML models, the former premises the same parameters for all respondents, while ML assumes that individuals have different preferences. Further, willingness-to-pay (WTP) was estimated when examining the monetary values of ESSs. WTP measures are useful for interpreting changes in a given attribute, and are calculated as follows:

$$WTP_k = -\frac{\beta_k}{\beta_{tax}}, \tag{10}$$

where $\beta_k$ is the parameter of attribute $k$, and $\beta_{tax}$ is the parameter of the tax.

## 3. Results

Table 4 lists the variables and definitions used in the analysis. Table 5 summarizes the estimation results of the CL and ML models. Attributes, except for "bird" and "recreations", were found to be statistically significant at the 1% level in the CL model, while the ML model showed statistical significance at the 1% and 5% levels except for recreations. As previously mentioned, the CL model assumes the same parameters for all respondents, leading to the overestimation in WTP. Similarly, the results demonstrate higher monetary values in CL compared to the ML model.

While exploring the heterogeneities in preference, remarkable findings can be observed in the ML estimation results. The normally distributed coefficients, estimated means, and standard deviations listed in Table 5 reflect the distribution of preferences. For example, the distribution of the seawalls for a 0.5 m~less than 1.0 m SLR coefficient had an estimated mean of 0.16, and an estimated standard deviation of 0.63, such that 62% of the distribution was above zero and 38% was below. This indicates nearly two-thirds of the respondents view a seawall for a moderate scenario as a positive and prefer it, whereas one-third do not prefer it. Similarly, seawalls for 1.0 m~less than 3.0 m SLR gained 76% positive responses with JPY 8846 in WTP, which was the highest percentage of all.

A remarkable finding regarding seawalls was that the standard deviations increased for seawalls as the extremity of scenarios increased, indicating significant heterogeneities in preferences. Regarding coastal forests, the results revealed that 58% of respondents preferred Eco-DRR, whereas 42% did not, and the average WTP/100 m of coastal forests was estimated to be JPY 695. Meanwhile, 58% of participants had negative preferences regarding an increased number of birds. In landscape attributes, gray + green (seawalls and coastal forests) obtained approximately 60% positive responses with JPY 3857 in WTP, while gray landscape (seawalls only) had 60% negative perceptions with a negative value of JPY −3852. However, in recreations, no significant results were obtained. To clarify those

heterogeneities in preferences, Figure 4 depicts the WTP distributions of each attribute. It is worth noting that the distribution of seawalls while preparing for extreme scenarios reflected the heterogeneities in preferences in comparison to others (Figure 4d).

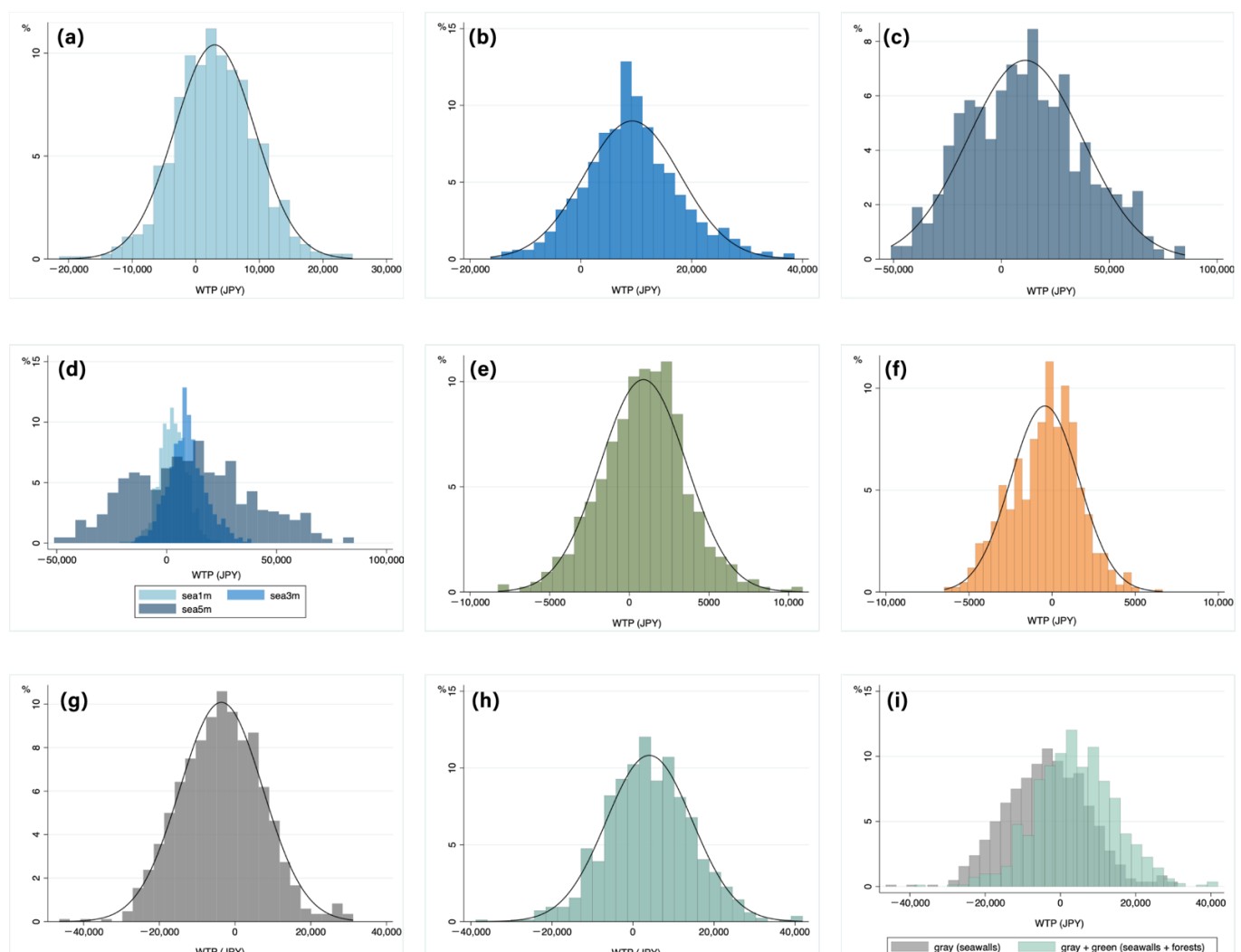

**Figure 4.** Distribution of willingness-to-pay (WTP). (**a**) Seawalls against SLR: 0.5 m~less than 1.0 m, (**b**) seawalls against SLR: 1.0 m~less than 3.0 m, and (**c**) seawalls against SLR: 3.0 m~less than 5.0 m. (**d**) Graph with integrated (**a**–**c**). (**e**) Coastal forests, (**f**) species richness, (**g**) gray landscape, and (**h**) hybrid (gray and green) landscape. (**i**) Graph with integrated (**g**,**h**).

**Table 4.** Variables and definitions.

| Variables | Definitions |
| --- | --- |
| asc | a dummy variable representing respondents' answers for alternative 4 (choose nothing) |
| sea1m | whether alternatives including sea-level is likely to rise ranged 0.5 m~less than 1.0 m were chosen (0–1 dummy) |
| sea3m | whether alternatives including sea-level is likely to rise ranged 1.0 m~less than 3.0 m were chosen (0–1 dummy) |
| sea5m | whether alternatives including sea-level is likely to rise ranged 3.0 m~less than 5.0 m were chosen (0–1 dummy) |
| forest | coastal forest width |
| bird | species richness (the number of avian species) |
| landsc_sea | whether alternatives including gray (seawalls) landscape were chosen (0–1 dummy) |
| landsc_both | whether alternatives including hybrid (seawalls and coastal forests) landscape were chosen (0–1 dummy) |
| rec_walk | whether alternatives including walking were chosen (0–1 dummy) |
| rec_fish | whether alternatives including fishing were chosen (0–1 dummy) |
| rec_camp | whether alternatives including camping were chosen (0–1 dummy) |

**Table 5.** Estimation Results of CL and ML models.

| | CL | | | ML | | | | |
|---|---|---|---|---|---|---|---|---|
| | | | | **Mean** | | **Standard Deviation** | | |
| | Coef. (s.e.) | | WTP (JPY) | Coef. (s.e.) | | Coef. (s.e.) | | WTP (JPY) |
| sea1m | 0.1669 | (0.06) *** | 3872 | 0.1555 | (0.07) ** | 0.6254 | (0.11) *** | 3186 |
| sea3m | 0.4008 | (0.07) *** | 9299 | 0.4317 | (0.09) *** | 0.8165 | (0.11) *** | 8846 |
| sea5m | 0.5498 | (0.04) *** | 12,756 | 0.5488 | (0.08) *** | 1.6381 | (0.08) *** | 11246 |
| forest | 0.0271 | (0.01) *** | 629 | 0.0339 | (0.01) ** | 0.2096 | (0.02) *** | 695 |
| bird | 0.0035 | (0) | 81 | −0.0189 | (0.01) *** | 0.1223 | (0.01) *** | −387 |
| landsc_sea | −0.1625 | (0.04) *** | −3770 | −0.1880 | (0.06) *** | 0.8348 | (0.07) *** | −3852 |
| landsc_both | 0.1273 | (0.04) *** | 2954 | 0.1882 | (0.06) *** | 0.8227 | (0.07) *** | 3857 |
| rec_walk | 0.1204 | (0.05) *** | 2794 | 0.1023 | (0.07) | 0.5468 | (0.08) *** | 2096 |
| rec_fish | 0.0235 | (0.04) | 545 | −0.0102 | (0.05) | 0.3779 | (0.1) *** | −209 |
| rec_camp | −0.0207 | (0.05) | −480 | −0.0434 | (0.06) | 0.4947 | (0.1) *** | −889 |
| price | −0.0431 | (0) *** | | −0.0488 | (0) *** | | | |
| asc | −0.2930 | (0.07) *** | | −0.7353 | (0.09) *** | | | |
| Number of obs. | 26,880 | | | 26,880 | | | | |
| Log likelihood | −8880 | | | −7870 | | | | |
| Pseudo R2 | 0.047 | | | 0.053 | | | | |

Note: ** significance at 5% level and *** significance at 1% level.

## 4. Discussion

This study explored coastal citizens' preferences for coastal infrastructure against future SLR, and other issues, such as how people value coastal ESSs, by considering the tradeoff relationship between them, the monetary values of coastal functions, and the perceptions of long-term settings using the stated preference method. For this discussion, the restrictions of this study must be determined.

First, this research assumed that the effects of SLR were consistent in any coastal region and incorporated it by increasing the initial water depth. Choice experiments seem to be suitable for estimating the monetary value of coastal ESSs because of the direct and indirect benefits for the people. Though choice experiments enable the exclusion of unrealistic coastal settings, the hypothetical settings of the questions asked inevitably leads to some kind of bias, or results in so much "noise" that the data are not useful for drawing inferences [49]. Thus, the stated preference method can be criticized for its validity. Additionally, how researchers provide the visual aids in choice experiments is likely to affect respondents' decision making.

The results in the ML model (Table 5) demonstrated that there was no significance in recreational attributes, which may reflect that recreation and landscape are intertwined, and the questionnaires design must address how the explanations regarding attributes should be presented to quantify their values more accurately. ESSs are combined with each other from an ecological point of view. Hence, it is crucial to improve the experimental design to consider these obstacles. During the choice experiments, respondents chose the most preferable coastal setting, which means that they agree to pay for it via an annual tax instead of using the same amount of value for other goods and services, however, the linkage between behaviors and statements has been controversial in stated preference methods from a psychological perspective [56].

Second, this paper began by defining seawalls as gray infrastructure and coastal pine forests as green infrastructure; thus, hybrid infrastructure was recognized as seawalls combined with coastal forests, although this is the first application of choice experiments to the valuation of coastal gray, green, and hybrid infrastructure, which focuses on seawalls and coastal forests, coastal ecosystem functions commensurate with other vegetation (i.e., sand dunes and coral reefs), as well as the diverse structural improvements in coastal armoring. Thus, it is inevitable to consider other forms of coastal engineering and natural components, and their integration in both onshore and offshore environments.

Third, ML is used to capture the heterogeneities in preferences, and has a more realistic substitution pattern than CL. However, it is necessary to take current coastal land use (i.e., residential districts, tourist destinations, seawalls, or coastal forests) and topography, such as mountainous coasts, into account. This study assumed that seawalls with a height of 2 m and 2000 m have been already constructed, however, the height and forms of seawalls varies in each place in a real world. Hence, it is worth considering how existed gray structures near respondents' residences affect respondents' decision making as a future work. In the coastal analysis, the low-elevation coastal zone (LECZ), where the contiguous area along the coast that is less than 10 m above sea level, has been described [4]. Following this coastal distribution, 33% of the respondents were associated with the LECZ. Further, considering the distance from coastlines, 18% of the respondents resided in coastal regions that are 10 m below the elevation and within less than 5 km from the coastlines (Table A1 in Appendix A). Thus, it is important to assess the extent to which low-elevation coastal populations are at risk of SLR, stronger storms, and other seaward hazards induced by climate change ([4]), and what individual characteristics affect their coastal infrastructural preferences using the latent class model. In addition, future works need to monitor the disparity of settlements between low- and high-income residents [57]. In tackling these issues, revealed preference methods such as the hedonic pricing method, which is one of the revealed preferences in non-market valuations and assumes that house prices might be affected by environmental attributes and neighborhood environments, will be useful. They usually focus on coastal erosion, beaches, and their impacts on recreation and tourism [58–62]. For instance, coastal residents in Mexico placed their importance on proximity to waterfronts as one of the attributes when deciding a settlement and higher prices were paid for houses located near most waterfront types [63]. The results in the ML model (Table 5) estimated the WTP of coastal forest per 100 m to be JPY 695, with positive attitudes of nearly 60%; however, enlarging coastal forests may cause an overlap with residences in some locations. Interestingly, there was no strong evidence that the presence of seawalls affected housing prices [64]. This obviously varies according to countries, regions, and their topographical characteristics. A recent study on spatial hedonic analysis revealed that the regional heterogeneities in hotel room pricing are affected by environmental and geographical attributes, such as urban areas, elevation, and distance from coasts [65]; however, whether coastal armoring and other environmental features are likely to affect housing values needs to be further explored. Therefore, analyzing using both revealed preference and stated preference methods will facilitate urban forestry in coastal zones by balancing gray and green infrastructures, daily uses, and disaster management from human dimensions, as well as provide some hints for sustainable coastal development, benefiting both humans and nature.

## 5. Conclusions

The increasing frequency and magnitude of natural hazards coupled with SLR have exacerbated the risks of coastal zones. This study quantified people's preferences using the ML model through choice experiments. Although this study has several constraints, including the validity of responses in stated preference methods, the scenarios of SLR, and their uncertainties, the results represent significant heterogeneity in the preferences of coastal citizens. While the coastal people in Japan acknowledged gray's coastal defense function, the diverse perceptions toward seawalls for intensifying against SLR were notable as its scenarios became severe. In addition, regarding landscapes, citizens showed positive attitudes toward hybrid landscapes compared with the negative WTP of gray-based coastal landscapes. Furthermore, coastal citizens' perceptions regarding Eco-DRR offered by coastal forests were positive, even though there are uncertainties in their performances.

**Funding:** This work was supported by the Environment Research and Technology Development Fund JPMEERF20184005 of the Environmental Restoration and Conservation Agency of Japan and JSPS KAKENHI Grant Number 19H04337.

**Institutional Review Board Statement:** Not applicable.

**Informed Consent Statement:** Not applicable.

**Data Availability Statement:** The data presented in this study are available from the corresponding author. The data are not publicly available due to privacy protection.

**Acknowledgments:** The author is deeply grateful for valuable comments on analysis and discussion with Koichi Kuriyama. The author also thanks Futoshi Nakamura, Takahiro Tsuge, Ayumi Onuma, Yasushi Shoji for their support and advice, as well as the research group from Tokushima University, Takumi Tsuruma, and Makoto Nakata from Niigata University for providing photos and comments on the questionnaire design. Furthermore, I received extensive support from Nikkei Research Inc., and would also like to thank the respondents of the online surveys.

**Conflicts of Interest:** The author declares no conflict of interest.

## Appendix A

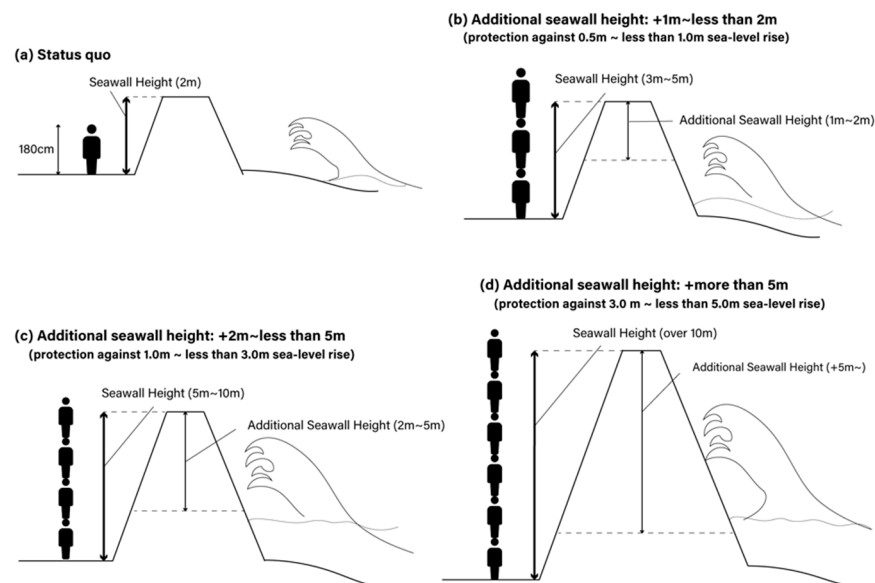

**Figure A1.** Additional seawall height: (**a**) status quo (seawalls with a height of 2 m and 2000 m have been already existed); (**b**) seawalls against SLR: 0.5 m~less than 1.0 m (the total seawall height is approximately 3 m~5 m by adding 1 m~2 m in height); (**c**) seawalls against SLR: 1.0 m~less than 3.0 m (the total seawall height is approximately 5 m~10 m by adding 2 m~5 m in height); (**d**) seawalls against SLR: 3.0 m~less than 5.0 m (the total seawall height is over 10 m). Note: For explaining briefly, the figure here displays the similar image that was used in the questionnaire.

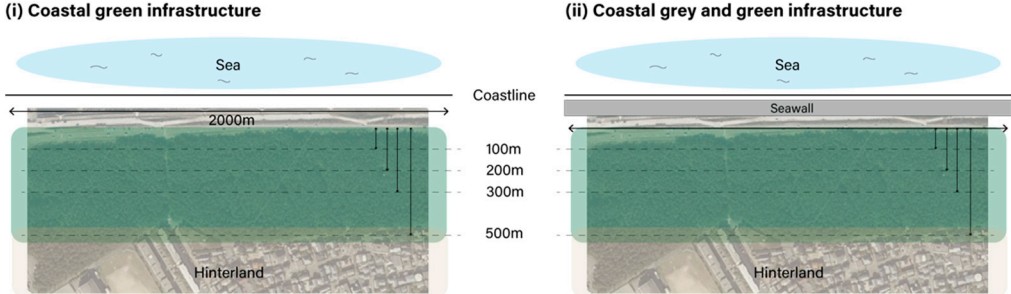

**Figure A2.** Coastal forest width. Coastal forest width is the distance from coastlines, and coastal forests were afforested behind the seawalls if they have been already constructed. Note: For explaining briefly, the figure here displays the similar image that was used in the questionnaire. Source: Geospatial Information Authority of Japan, edited by the author.

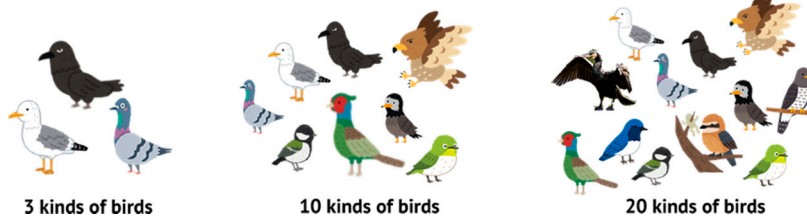

**Figure A3.** A number of avian species. The small number of bird species indicated that *Columba livia*, *Larus crassirostris*, and *Corvus* (*Corvus macrorhynchos* or *Corvus corone*), which are generally seen in Japan, can be observed while a wide range of avian species is described as raptors and includes *Pandion haliaetus* and *Accipiter gentilis*, which are associated with higher consumers.

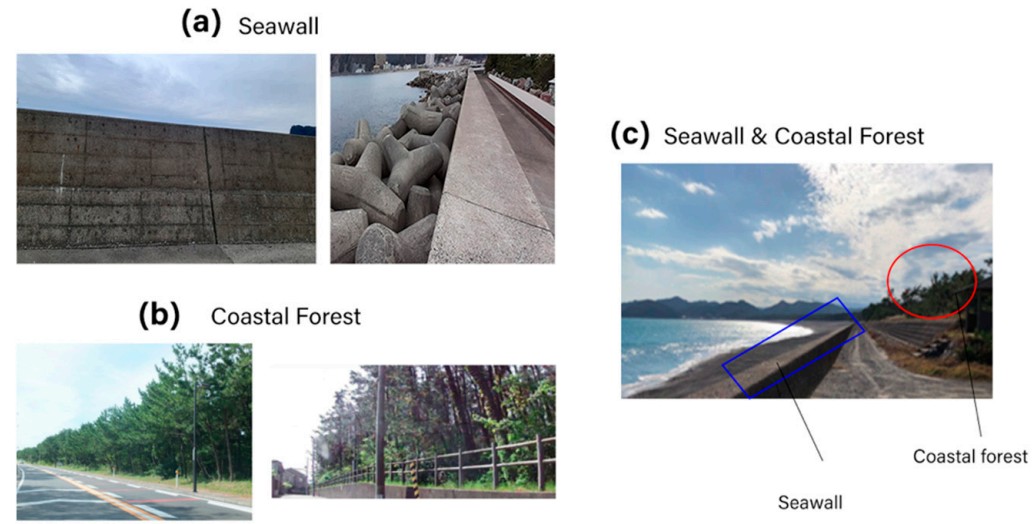

**Figure A4.** Landscape: (**a**) seawalls (gray), (**b**) coastal forests, and (**c**) seawalls and coastal forests (gray and green: hybrid).

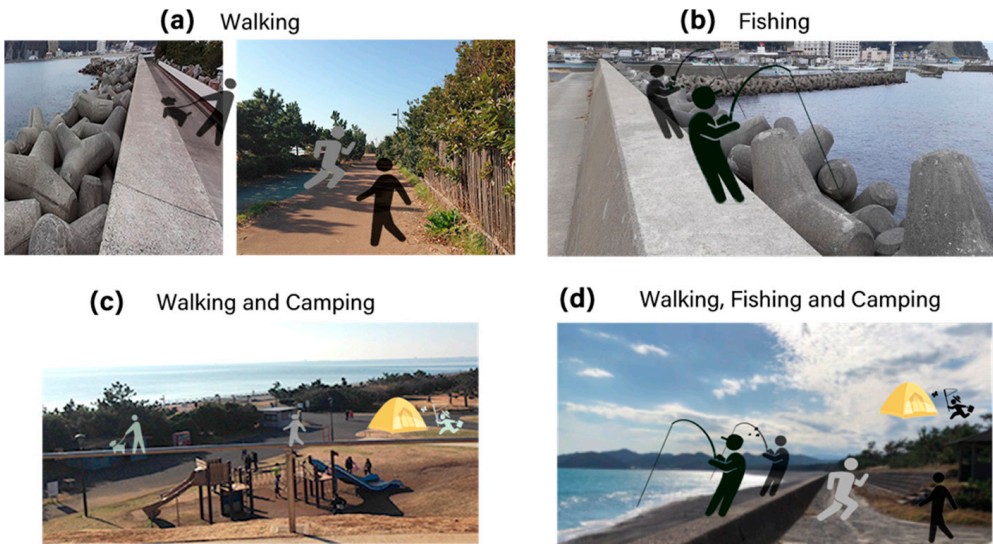

**Figure A5.** Recreation: (**a**) a promenade provided for walking along coastlines, (**b**) fishing is permitted, (**c**) a promenade for walking and a space for camping near the ocean, and (**d**) all three activities (**a**–**c**) are available.

**Table A1.** Lower Elevation Coastal Zone in Japan.

| Classification | | Definition | Data |
|---|---|---|---|
| LECZ | Near coasts | 10 m below in elevation and within less than 5 km from the coastlines | 147 (17.5%) |
| | Away from coast | 10 m below in elevation and over 5 km from the coastlines | 127 (15.1%) |
| Non-LECZ | Near coasts | 10 m above in elevation and within less than 5 km from the coastlines | 111 (13.2%) |
| | Away from coast | 10 m above in elevation and over 5 km from the coastlines | 455 (54.2%) |

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
