# Peer review of "Preference Heterogeneity of Coastal Gray, Green, and Hybrid Infrastructure against Sea-Level Rise: A Choice Experiment Application in Japan"

_sustainability, doi:10.3390/su13168927_

Round 1

Reviewer 1 Report

Please find attached the comments.

Reviewer 2 Report

Dear authors, 

I thank you for giving me the possibility to review the paper named "Preference heterogeneity of coastal gray, green, and hybrid in
frastructure against sea-level rise: A choice experiment applica
tion in Japan". I hope you and your family are safe and well.

I read twice your paper and I find it interesting. However I find out that some improvements are required :

a) You often speak about resilience. But I don't find a description or a definition what you mean in your case. I find out you have to define it and adapt to your study

b) About Simulation, why do you think to use dicrete model?I don't understand. I think you should consider dynamic model because the sea-rise depends on different factors that can be interralated. Look at this paper :  https://doi.org/10.15866/iremos.v9i4.9688 . The authors consider in safety management the dynamic approach. PLease consider it and improve your paper. I suggest only this, but they also reaserched about resilience and so on. But looking at your references, I find you studied it, even if there is no formulation. 

c) about conclusions  and discussions.  What is the real weakness of your model? What will be the research in future?

Please improve your paper, and you will get the result.

Author Response

This manuscript is a resubmission of an earlier submission. The following is a list of the peer review reports and author responses from that submission.